# Comparing global trends in marine cold spells and marine heatwaves using reprocessed satellite data

**Robert Peal, Mark Worsfold, and Simon Good**

Met Office, FitzRoy Road, Exeter, Devon, EX1 3PB, UK

**Correspondence:** Mark Worsfold (mark.worsfold@metoffice.gov.uk)

**Abstract.** Climate change is causing extreme climate events to become more frequent and more severe. Marine heatwaves (MHWs) and marine cold spells (MCSs) are prolonged, discrete periods of anomalously high or low ocean temperatures with wide-ranging impacts from dramatic shifts in biodiversity to changes in fishery yields. Previous research has found that MHWs are increasing in frequency and intensity, but MCSs remain less well understood.

We used sea surface temperature (SST) data to compare the global observed MCS and MHW intensities and trends in their frequency over the period 1982–2021. These events were also assigned a category from I (moderate) to IV (extreme). Our findings show that in large areas of the ocean it can be said with 99 % confidence that MCSs have become less frequent and MHWs have become more frequent. In those regions, the occurrence of MCSs has typically reduced by one event every 5 years, while there is one extra MHW every 5 to 10 years. However, parts of the Southern Ocean go against these trends with MHWs becoming slightly less frequent and MCSs becoming more frequent.

The trend of increasing numbers of MHWs and decreasing numbers of MCSs is mostly due to increases (and decreases) in category I and II events. Category III and IV MHWs are less common than the milder events, and a trend analysis demonstrated that across most of the ocean they are occurring at a slightly increased rate. However, the spatial extent of the ocean affected by these events is increasing at a faster rate. The occurrences of category III and IV MCSs are infrequent, and a trend analysis also found that their rates of occurrence, as well as the spatial extent of the ocean affected, remain near constant.

Many of our results are in agreement with previous studies; what is significant about this study is the fact that it uses a different, higher-resolution input SST dataset. The similarity of the results between the different research efforts strengthens the argument that these events are not just a feature of their input dataset.

## 1 Introduction

There is now overwhelming evidence that climate change will increase the frequency and intensity of extreme climate events such as marine heatwaves (MHWs) (IPCC, 2021; Frölicher et al., 2018; Oliver, 2019), which are defined as "prolonged discrete anomalously warm water events" (Hobday et al., 2016) and have attracted a growing amount of interest in recent years. These events often have dramatic and sometimes devastating impacts on local ecology including shifting local biodiversity (Wernberg et al., 2013, 2016; Frölicher and Laufkötter, 2018) and triggering coral bleaching events (NOAA, 2015). They also have a range of socio-economic impacts (Smith et al., 2021) such as altering fishery yields (Mills et al., 2013; Smale et al., 2019; Cheung et al., 2021). Marine cold spells (MCSs), which are defined equivalently to MHWs as a "discrete, prolonged anomalously cold water event at a particular location" (Schlegel et al., 2021), remain less well understood, but a growing body of research suggests that low-SST (sea surface temperature) events can also have significant impacts on ecology including coral reef destruction (Lirman et al., 2011) and have links to extreme climate events on land (Duchez et al., 2016). Previous re-

**Table 1.** Data used for report.

| Product ref. no. | Product ID and type | Data access | Documentation |
|---|---|---|---|
| 1 | SST_GLO_SST_L4_REP_OBSERVATIONS_ 010_011; satellite observations | EU Copernicus Marine Service Product (2022) | Quality Information Document (QUID): Worsfold et al. (2022a) Product User Manual (PUM): Worsfold et al. (2022b) |

search into MCSs has focussed on certain regions including investigating drivers of shallow-water events in some coastal regions (Schlegel et al., 2017) and effects of MCSs on fisheries in the North Sea (Wakelin et al., 2021).

The field of MHW and MCS research moves rapidly, but two research papers in particular are very relevant to this investigation. Both papers used the Optimum Interpolation Sea Surface Temperature (OISST) climate dataset produced by the National Oceanic and Atmospheric Administration (NOAA) and the same detection method (Sect. 2) over the period 1982–2020.

Schlegel et al. (2021) provided a comprehensive review of previous studies into MCSs and the possible causes of MCSs, which were found to be often driven by anomalous winds, although other drivers were also identified. The trends in MCSs were also investigated for different MCS categories. Wang et al. (2022) calculated and compared the trends in MHWs and MCSs but did not delve into event categories.

This investigation has some parallels with the first two, but it is the first to use the higher-resolution Operational Sea Surface Temperature and Ice Analysis (OSTIA) climate SST dataset (Table 1) produced for the Copernicus Marine Environment Monitoring Service (CMEMS), and it can therefore provide an independent assessment of the occurrence of MHWs and MCSs.

## 2 Methods

### 2.1 Input SST dataset used

Most previous MHW–MCS investigations have used the 0.25° resolution NOAA OISST climate SST (version 2.1) dataset (for example Schlegel et al., 2021, and Wang et al., 2022), which represents the SST at 0.5 m depth (Huang et al., 2020). This is often called a SST depth product, which will increase during the day due to solar heating and cool at night (termed a diurnal cycle).

This investigation used the 0.05° resolution CMEMS OSTIA climate dataset for the period 1982–2021 (Table 1). OSTIA is formed from nighttime SSTs, and daytime SSTs whenever windy conditions mean that it can be assumed that the diurnal cycle is not affecting the daytime temperature (Good et al., 2020). Therefore, it represents the foundation SST, which can be defined as the temperature from which the growth and decay of the diurnal heating develops each day (GHRSST, 2022).

OSTIA uses the NEMOVAR variational data assimilation scheme (Mogensen et al., 2009) to combine satellite and in situ data with a background field that is based on the previous day's analysis (Good et al., 2020). In areas of the ocean where there are no observations, if the sea ice concentration is greater than 50 %, the SSTs will eventually relax to $-1.8\,°C$ and elsewhere to a 1985–2007 climatology.

As a result of RAM memory overflow issues encountered by the R programming language (unfortunately a fundamental limitation of this language) when processing the full-resolution data, the OSTIA dataset was linearly regridded to 0.25° resolution for the purpose of this investigation.

### 2.2 Definition and detection of marine heatwaves and marine cold spells

Fundamentally an MHW is detected whenever the SSTs exceed a threshold value, but this threshold has a variety of interpretations depending on the investigation requirements. One of the simpler definitions is to use a chosen percentile (usually 99th) as a threshold SST value calculated from a chosen climatology period (Darmaraki et al., 2019; Frölicher et al., 2018; Laufkötter et al., 2020). For this study we are using the Hobday definition of a heatwave (Hobday et al., 2016, 2018; Schlegel et al., 2021; Wang et al., 2022), which has gained wide acceptance in the MHW–MCS research community. To detect an MCS, consider that each pixel of the dataset has the following mathematical elements:

A. a time series of SST values (thin black line in Fig. 1a and b)

B. the daily climatology using the period January 1982–December 2011 as the baseline (smooth, thicker black line in Fig. 1a and b)

C. the 10th percentile of the climatology data $B$, taken from an 11 d rolling window centred on each day (solid green line in Fig. 1a and b), which is the threshold used to determine if there is an MCS

D. the difference between the climatology $B$ and the 10th percentile $C$.

A pixel is considered to have an MCS wherever the SST value $A$ is below the 10th percentile threshold value $C$ for at least 5 d. If there is a gap of less than 2 d between an MCS, they are considered to be one event.

Furthermore, the difference value $D$ then allows this MCS to then be categorized. A heatwave is considered category I (moderate) when the SST value $A$ is less than the sum of the climatology value $B$ and the difference value $D$ (i.e. where $A <= B + D$) (below solid green line in Fig. 1a and b). A category II (strong) MCS would be where $A <= B + 2D$ and so on for severe and extreme events (dashed green lines in Fig. 1a and b).

MHWs are defined equivalently as a period of at least 5 d where the SST value is above the 90th percentile of the SST data from an 11 d rolling window centred on each day.

Using the Hobday definition, events can be described by the following characteristics:

– Duration is the number of consecutive days where the SST exceeded the 90th percentile value for MHWs or was below the 10th percentile value for MCSs.

– Maximum intensity is the maximum difference between the SST and the climatology value during the event (MHWs have positive intensity, and MCSs have negative intensity).

– Category describes the severity of an event by assigning a category to the maximum intensity value relative to the local climatology.

All these MHW and MCS characteristics are calculated for each individual grid cell separately; the neighbouring cells are not considered.

The MHWs and MCSs and their characteristics were calculated from the SST dataset using the heatwaveR package (Schlegel and Smit, 2018; Hobday et al., 2016). An important consideration is that areas with persistently high SSTs do not necessarily suffer as many heatwaves. While this may seem counter-intuitive at first, consider that MHWs are defined relative to the climatology. Therefore, an area with historically high average SST values will present less opportunity for recent SSTs to exceed that high climatology threshold. This characteristic is valuable since researchers are often most interested in areas with significant deviations from historical temperatures.

# 3 Results and discussions

## 3.1 Using the Hobday MCS framework to describe anomalous low-SST events

To highlight the consequences of cold spells and how the Hobday category framework (Hobday et al., 2016, 2018)

can help describe the severity of MCSs, two case studies of known cold-water events were selected and investigated.

The first case study illustrates the effect of MCSs on marine ecosystems (Fig. 1a). An investigation by Abram et al. (2003) linked the coral reef death off Sumatra in October 1997 to the combination of Indonesian wildfires exacerbated by El Niño and anomalously low SST caused by the Indian Ocean Dipole (IOD) which caused increased upwelling of cold, nutrient-rich water in the region. Such upwelling events are associated with increased biological activity, which can lead to large algal blooms (often called "red tides") (Genin et al., 1995). These events can limit oxygen supplies to coral reefs but usually pose only a minor threat as the biological activity is usually iron limited (Johnson et al., 2001). However, in October 1997 the ash cloud from the Indonesian forest fires dumped large quantities of iron on the ocean, causing one of the largest red tides the region had ever seen, leading to mass asphyxiation of the coral (Abram et al., 2003).

This event was examined using the Hobday framework, and Fig. 1a shows that the anomalous SST in the region may be classified as a category IV MCS with a duration of around 100 d peaking in early November.

Extreme low-SST events have also been linked to significant global weather events, which is illustrated by the second case study (Fig. 1b). Duchez et al. (2016) showed that the 2015 European summer continental heatwave, 1 of the top 10 hottest in the last 65 years, as well as several other severe European summer heatwaves, were preceded by anomalously cold SSTs in the North Atlantic, but establishing causality (if any) is the subject of further research.

The inset in Fig. 1b shows that a category III MCS prevailed over much of the region in 2015, with some areas classified as a category IV. Further time series analysis of a pixel within a category IV region shows an MCS with a duration of around 69 d peaking in late March.

In these examples, the Hobday category framework has been used to describe the severity of marine SST anomalies that have been linked with significant environmental and ecological events. Use of this framework may help researchers to better communicate the severity of both low- and high-SST anomalies with public and local officials, allowing the area to prepare for the effects of marine temperature extremes.

## 3.2 A global perspective of MHW and MCS distribution and strength

The average maximum intensity of events and the average number of yearly events for both MHWs and MCSs were calculated over the global domain for the period 1982–2021 to produce a map of the distribution of such events (Fig. 2). Figure 2a and b show that the mean maximum intensity of both types of events at most locations in the global ocean was 1 to 2 °C and between 2 and 4 °C in the Niño 3 region of the eastern Pacific (defined as 5° N–5° S, 90–150° W) (Trenberth, 1997). Moreover, other regions with higher-intensity

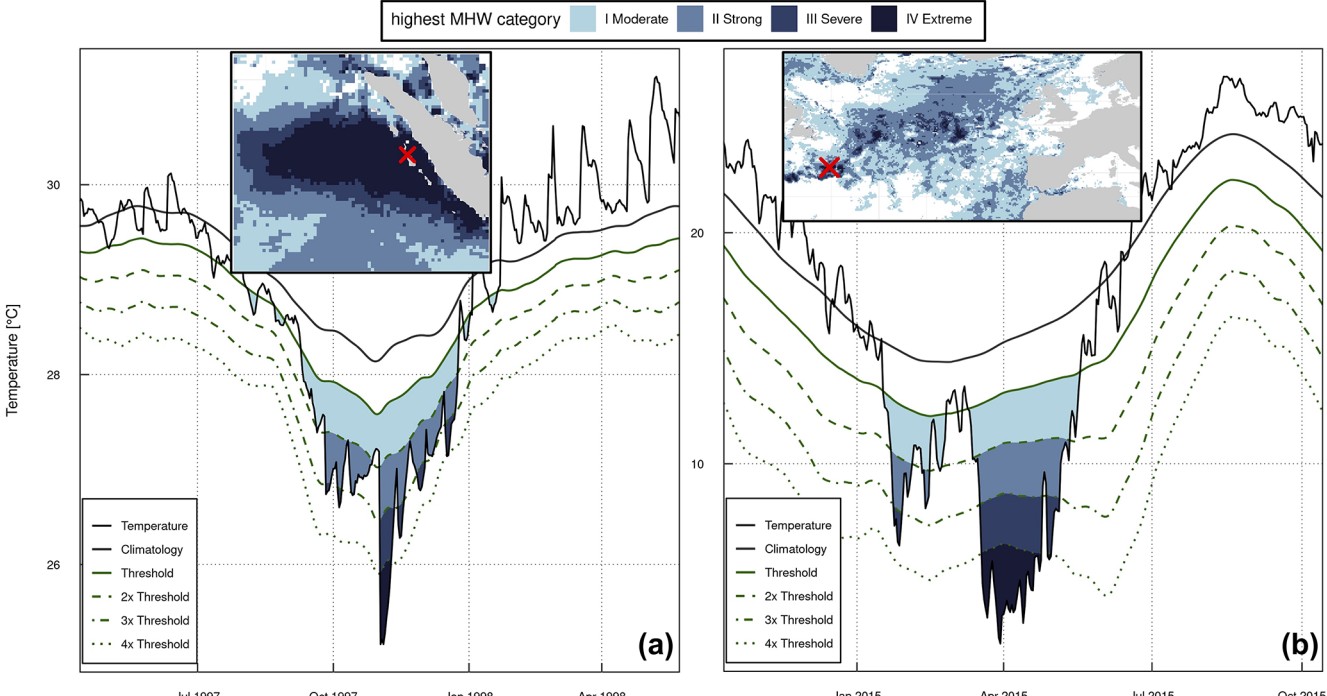

**Figure 1.** Significant MCSs. **(a)** Sumatra 1997. Main figure: SST values and MCS threshold during late 1997 at lat 0.875, long 98.625. Inset: highest MCS category detected across the region during 1997 (the red cross denotes the location where the time series data were extracted). **(b)** North Atlantic 2015. Main figure: SST values and MCS threshold during early 2015 at lat 41.125, long −50.375. Inset: highest MCS category detected across the region during 2015 (the red cross denotes the location where the time series data were extracted).

MHWs (sometimes exceeding 6 °C) also experienced MCSs of similar intensity. These regions of high-intensity MHWs–MCSs were also located in areas in which western boundary currents (WBCs) are found (NW Atlantic – Gulf Stream, SW Atlantic – Brazil Current, South Africa – Agulhas Current, Japan – Kuroshio Current; Le Traon and Morrow, 2001). WBCs are found at the western edges of ocean gyres and transport warmer waters towards the poles (Holbrook et al., 2019).

The average number of heatwaves per year was calculated for each pixel of the dataset and plotted. Examining Fig. 2c and d it can be seen that in most locations, MHWs were slightly more common than MCSs, with between one to three events detected per year compared to one to two MCS per year.

However, MHWs were rare in the Niño 3 region, occurring no more than once per year, while MCSs occurred between two and three times per year in the same region.

Regions with a higher frequency of MCSs tended to coincide with regions where they were more intense. The northwest Atlantic, south-west Atlantic, region off South Africa, and north-west Pacific off Japan all had up to three MCSs per year.

Schlegel et al. (2021) carried out a similar investigation into MCS count and intensity for the period 1982–2020, and Oliver et al. (2018) also did the same for MHWs for the 1982–2016 period. As mentioned earlier, Wang et al. (2022) compared the trends in MCSs and MHWs (using similar but different metrics). An important point to reiterate is that these previous studies had used the NOAA OISST dataset.

Despite the differences between investigations, it is remarkable that the spatial distribution of the values in Fig. 2 are very similar to the results from the previous investigations.

### 3.3 An exploration of linear trends in MHW and MCS events for each category

In order to determine whether there was a change in the number of MHWs and MCSs per year, a linear trend was calculated for each pixel of the global dataset and plotted (Fig. 3). Further plots were also generated for each event category, and contours were added wherever the trend exceeded a 99 % confidence level.

Figure 3a shows that occurrences of MCSs in much of the Atlantic, Indian, and western Pacific oceans are decreasing at a rate of up to one event every 5 years and that these decreases exceed the 99 % confidence interval for most of the region. Figure 3c shows a large decrease in category I (moderate) events at a rate of almost one event every 5 years at the locations which exceed the 99 % confidence value. A smaller but still significant decrease in category II (strong) events of

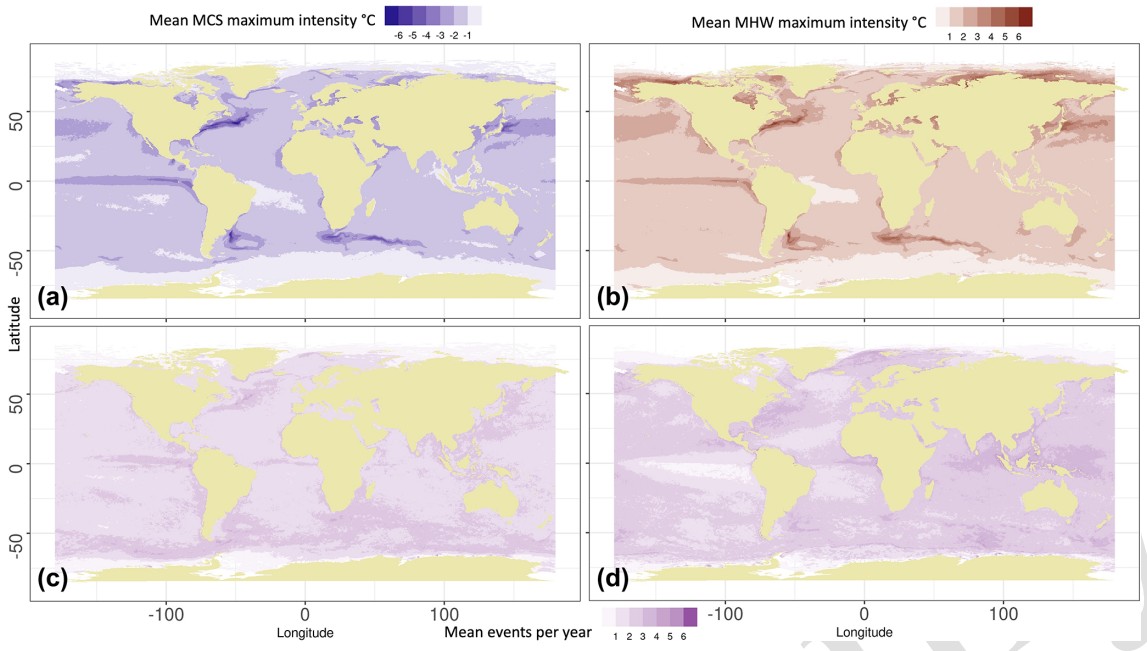

**Figure 2.** Comparison of global MHW and MCS mean maximum intensity and frequency of occurrence during the period 1982–2021, calculated from product 2.1.1. **(a)** Mean MCS maximum intensity and **(c)** average number of MCSs per year at each grid point. **(b)** Mean MHW maximum intensity and **(d)** average number of MHWs per year at each grid point.

about one event every 20 years was also observed (Fig. 3e). Figure 3g and i show that category III and IV events are continuing to occur at a roughly constant rate. Therefore, the overall decrease in MCS is largely due to decreases in the numbers of category I and II events.

Figure 3b shows that MHWs in the central Atlantic, Indian, and western Pacific oceans are increasing by one event every 5 to 10 years, and this trend is significant to the 99 % confidence level in much of the region. Figure 3d and f show that this is largely due to increases in category I and II MHWs of up to one event per 10 years in much of the region. Figure 3h and j show that the trends in category III and IV MHWs are close to zero.

It is also notable that the southern Pacific Ocean (in particular, east of South America) goes against the general trends. Figure 3a shows that MCSs are becoming more frequent in the southern Pacific Ocean by one event every 10–20 years, which is largely due to increases in category II events (Fig. 3e). Meanwhile MHWs are also becoming less frequent in this region by one event every 10–20 years (Fig. 3b), which is largely due to decreases in category I events (Fig. 3d).

Interestingly, the increasing number of MCSs (for all categories) in the Southern Ocean was also observed by Schlegel et al. (2021), although their research did not determine whether events of a specific category were responsible for this behaviour.

Oliver (2019) determined that rising average SSTs were the main driver of trends in MHWs, which was also con-

firmed by Wang et al. (2022), who also determined that rising SSTs were responsible for trends in MCSs. In addition, climate change research has also found that, while SSTs are increasing on average, the southern Antarctic Ocean has cooled slightly (Auger et al., 2021; Rye et al., 2020; Armour et al., 2016). The observed spatial patterns in our results for MHWs and MCSs fits into a wider picture of the ocean's behaviour changing as a consequence of climate change.

## 3.4 Changes in the affected ocean area over time

To investigate whether there are any changes in how much of the ocean is affected by MHWs and MCSs, each pixel was examined to determine the most severe event that had occurred in each year. This information was used to produce a time series for the fraction of the ocean area affected by an event (Fig. 4).

Figure 4 (left) shows that MCSs are now occurring in less of the ocean than historically. The amount of ocean where the highest MCS categories detected were category I and II has decreased from 41.8 % in 1982 to 25.3 % in 2021 and from 27.8 % in 1982 to 13.2 % in 2021 respectively, while the amount of ocean without any MCSs at all has increased from 26 % to 56.6 % in 2021. In contrast, category III and IV MCSs have been detected at a near constant ∼ 1 %–3 % of the ocean. This suggests that even in a warming world, extreme MCSs have so far remained consistently rare.

The investigation by Schlegel et al. (2021) also found that category III and IV events remained steady and that the area

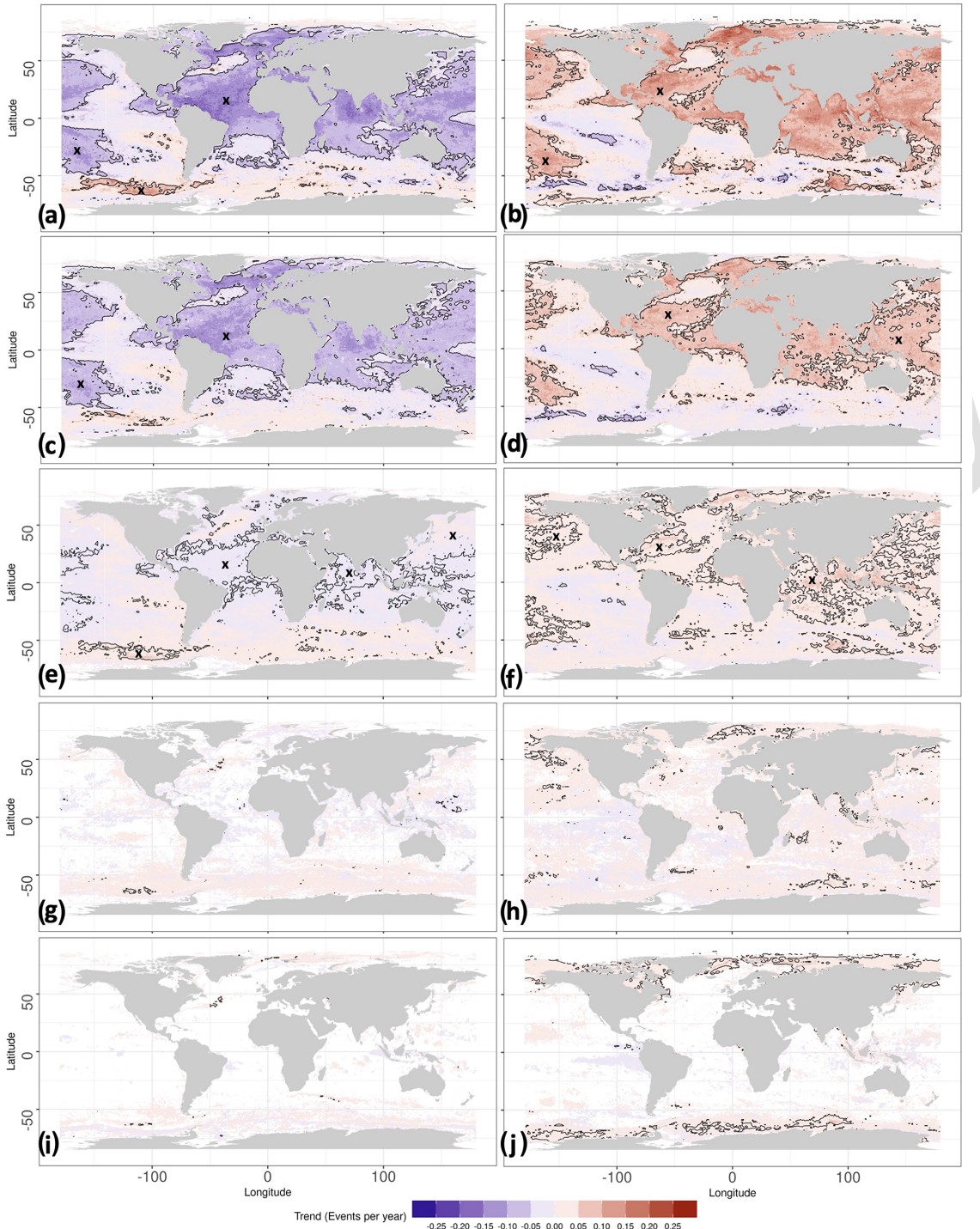

**Figure 3.** Trends in the number of MCS and MHW events per year for **(a)** all MCSs, **(b)** all MHWs **(c, e, g, i)**, category I–IV MCSs **(d, f, h, j)**, and category I–IV MHWs. The year of an event has been defined as the year of peak intensity. Linear trends were calculated using a least-squares fit, and the contour line bounds the region where the trends exceed the 99 % confidence level of a two-tailed $t$ test compared to a constant rate of occurrence. The X markers indicate the inside of the contour to aid interpretation. Areas with no shading are where no events were detected.

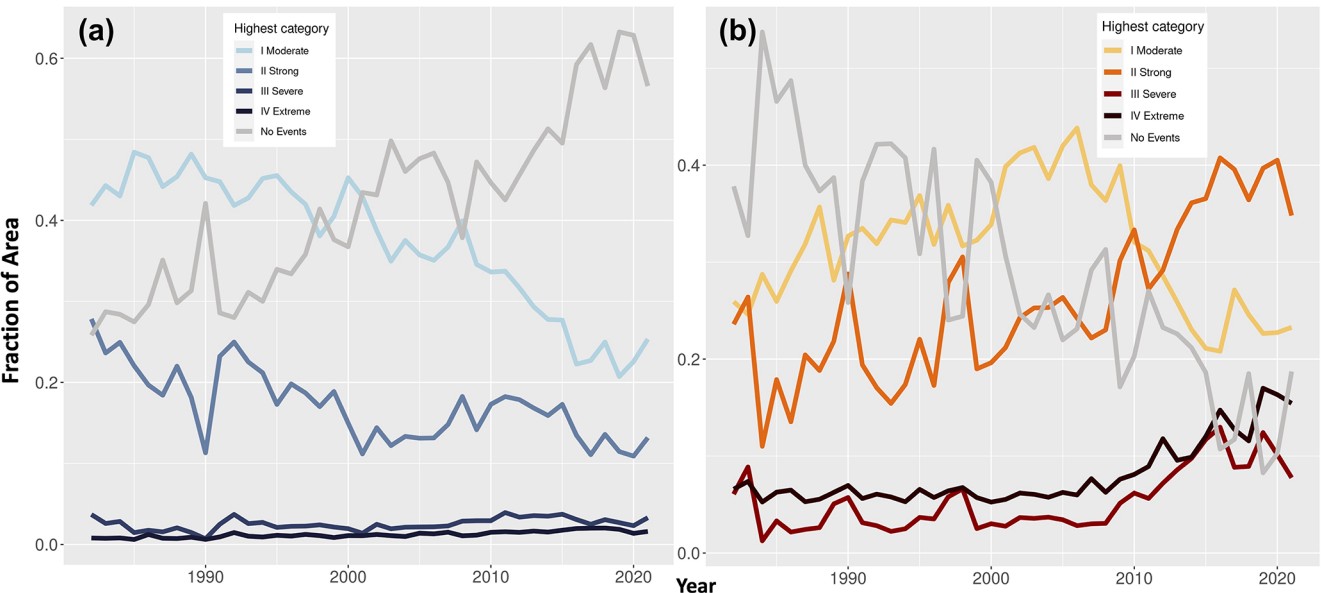

**Figure 4.** The fraction of the area of the ocean where the highest **(a)** MCS and **(b)** MHW category detected was I, II, III, IV, or no events being detected. The year of an event is defined as the year of the peak intensity.

of the ocean which experienced an MCS had decreased (61 % in 1982 to 25 % in 2020). However, their investigation differed in that they determined that this was mainly due to a decrease in category II–IV events.

Figure 4 (right) demonstrates that the area of ocean with no MHWs has seen a steady decline throughout the time period from 37.8 % in 1982 to 18.7 % in 2021, indicating that MHWs are occurring at more locations in the ocean. Breaking down the results by category, it can be observed that category I events were initially the most common; the area dominated by these events increased from 25.9 % to 43.8 % between 1982 and 2006 before decreasing sharply to just 23.3 % in 2021. In the same period, the area of ocean where II was the highest MHW category increased from 23.6 % to 34.8 % in 2021, overtaking category I as the most common category. The amount of ocean where the highest category was III or IV initially remained at around 3 % and 6 % respectively until 2008, after which there was a sharp increase in both categories to 7.7 % for category III and 15.4 % for category IV in 2021. The spatial increase in these categories is especially a cause for concern since these types of events tend to have more dramatic impacts compared to lower-category events (Hobday et al., 2018).

It is also interesting to note that the same investigation by Hobday et al. (2018) also produced a graph displaying a similar behaviour (a reduction in "no events" and category I MHWs and an increase in other categories).

## 4  Conclusions

In this study, we have compared global MHWs and MCSs in the period 1982–2021. We have revealed that the mean peak intensity of MHWs and MCSs at most locations was 1 to 2 °C and the mean peak intensity was highest (up to 6 °C for both kinds of event) in regions where western boundary currents dominate. We have also shown that in much of the ocean, MCSs have become less frequent by around one event every 5 years, while there is one extra MHW event every 5 to 10 years. Most of the changes are due to increases in category II–IV MHWs and decreases in category I and II MCSs. However, events in parts of the Southern Ocean go against these trends with MHWs becoming slightly less frequent and MCSs becoming more frequent. Category III and IV MHWs remain less common than the less severe events, and at individual locations they are occurring at a slightly increased rate over time. However, these events are becoming more widespread. Category III and IV MCSs are rare but are continuing to occur at near-constant rates in much of the ocean.

The average global frequency and maximum intensity (Fig. 2) and the trend analysis for all categories of events (Fig. 3a and b) had a similar spatial distribution to the MCS results obtained by Schlegel et al. (2021) and Wang et al. (2022) as well as the MHW results obtained by Oliver et al. (2018). The temporal changes in the ocean area affected (Fig. 4) were also similar to the results from investigations carried out by Schlegel et al. (2021) and Hobday et al. (2018).

The mechanisms that drive these trends in MHWs and MCSs are still under investigation, and further research would provide valuable insight into the changing nature of

MHWs and MCSs. Future research also should focus on improving understanding of how MCSs form and their impact on ocean ecosystems as well as on developing tools for predicting when they will occur, allowing for early warnings of these events. Although more research is required to quantify the differences in the outputs from the OISST and OSTIA datasets and their relative confidence levels, the similarity of the results strengthens the argument that these events are not just a feature of their input dataset.

It also demonstrates the potential of using OSTIA for future MHW–MCS investigations. Research carried out by Yang et al. (2021) showed that the full-resolution OSTIA was superior to OISST in resolving smaller oceanic features. Unfortunately, fundamental limitations in the R programming language prevented us from using the full-resolution OSTIA dataset in this investigation. However, this presents an excellent topic for a follow-up investigation using the full-resolution dataset (using a Python version of the MHW code) to determine whether this brings about improvements in MHW–MCS detection or delivers greater insights into their formation and behaviour.

**Data availability.** Publicly available datasets were used and are available from CMEMS (Table 1).

**Author contributions.** MW generated the initial datasets for the period 1982–2020 required for the investigation. RP processed the data to produce the results outlined above and wrote the first draft of the report. When data for 2021 became available, MW then updated the results with the latest data and performed the necessary rewrites for the review and submission process. SG also reviewed and edited the report.

**Competing interests.** The contact author has declared that none of the authors has any competing interests.

**Disclaimer.** Publisher's note: Copernicus Publications remains neutral with regard to jurisdictional claims in published maps and institutional affiliations.

**Acknowledgements.** The authors would like to acknowledge the following contributors: Robert Schlegel and Albertus Smit for their support with adapting the heatwaveR package to process global data and more generally making their excellent software available to the wider research community so that we can all benefit and Owen Embury of Reading University who made available his verification software, which was used in order to check that the code used to regrid the OSTIA dataset did not degrade the quality of the data. The OSTIA and OISST datasets were compared to independent in situ SST measurements and found to be similar. The results of this investigation were not included in this report since it was a quality control check rather than a research investigation. More information can be obtained from the authors on request.

**Financial support.** This work was partly funded by the CMEMS SST Thematic Assembly Centre (21001-COP-TAC Lot 3).

**Review statement.** This paper was edited by Johannes Karstensen and reviewed by two anonymous referees.

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
