# Peer review of "Comparing Global Trends in Marine Cold Spells and Marine Heatwaves using reprocessed satellite data"

_State of the Planet, 2022_

## Referee Comment (RC2)

**Review: "Comparing Global Trends in Marine Cold Spells and Marine Heatwaves" by Robert Peal et al.**

This manuscript presents statistics and trends of cold extreme events, namely Marine Cold Spells (MCS), using the high-resolution CMEMS OSTIA sea surface temperature product, which is a foundation SST product, that is free of diurnal variability. The authors present two case-studies to demonstrate the application of the MCS detection algorithm. Then global comparison between MCS and Marine Heatwaves (MHW) statistics including trends is presented.

Ocean temperature extreme events, in particular MHWs, have received a lot of research attention over the past years, mainly due to their potentially devastating impacts on marine ecosystems as well as socio-economic impacts, through e.g. fisheries.

While the manuscript is logically structured, I have concerns as to the novelty of the results presented and recommend to re-consider publication of this manuscript after major revisions. Detailed comments are listed below.

**Main comments**

The authors claim that they are the first study to examine trends MCS and additionally comparing these to trends in MHWs. This is not true, as this study https://agupubs.onlinelibrary.wiley.com/doi/full/10.1029/2021GL097002, published in March 2022, does exactly that and the results presented by Peal et al in Fig.2 and 3 are basically identical with Fig.1 and Fig.2 in Wang et al., 2022. The authors of the submitted manuscript do not cite this publication, which is a major flaw. I therefore recommend that the authors have to revise their manuscript significantly to demonstrate the novelty (if there is any) of their results.

Furthermore, the authors mention that one novelty is using the CMEMS OSTIA dataset instead of OISST. While I see some value in comparing results using different datasets, I feel like the authors should then use chance to provide more quantitative details into the differences, e.g. what actual difference does it make for MHW/MCS statistics when I used OISST vs. OSTIA? How does this differ from region to region? This comes down to the drivers of the events, where do I really need such high resolution and where is 0.25° sufficient? Does OSTIA provide better results in coastal regions or areas of high SST gradients, such as Western Boundary Currents? Also the authors state that they are regridding OSTIA to 0.25° due to computational reasons. How does that affect the results? The authors could try to quantify this on a regional example if memory is an issue for the global dataset. I am not an R user but there should still be ways around these memory issues by e.g. applying the detection code on lat/lon chunks of the data. In addition, highlight the differences of using a bulk vs. a foundation product? How does that impact my MHW/MCS statistics?

If this manuscript is intended to be more of a report on the current state, the authors should better discuss outstanding issues, challenges and also highlight potential drivers in their discussion. A main part though is being up to date on existing literature.

**Minor comments**

There is little to no new insight provided as to the drivers of these events. If not through analyses, then the trend patterns should at least be discussed in the context of existing literature on temperature trends and their drivers.

Section 3.4. provides some new insights but is again lacking some interpretation.

**Figures**

Fig 3: Why use such a pale colormap? Panels e)-j) are very hard to see. The similar comment applies to Fig2 panels c) and d). I believe using a sequential colormap with more colors would help the readability of the figures.

Fig 4: I suggest making the lines a little thicker for better readability.

**Line-based comments**

L27: add 'can' after (MCSs), since not all events have devastating impacts

L42: same point as in L27. Change to for example 'These events often have dramatic…'

L47: Site https://www.science.org/doi/10.1126/science.abj3593 for socio-economic impacts. Change sentence to e.g. "… having socio-economic impacts (Smith et al., 2021) by e.g. altering fishery yields (Mills…)"

L78ff: Schlegel et al., 2021 should be clearly mention somewhere here.

L86: Why not use the term commonly used 'threshold'? The categories are then one,two,three,four times the threshold as also presented in Fig 1. This would seem more intuitive than presented here.

L93: Add that if there is a gap of less than two days between MHWs they are considered as one event.

L134ff: It would be more intuitive (from an oceanographer's point of view) to state at the beginning that Western Boundary Current regions show higher intensity and then list the regions.

L171-176: There are several ocean papers that discuss cooling trends over the Southern Ocean, these could be mentioned here to at least provide the reader with insights as to potential mechanisms behind these trends. Hence it is not surprising that Schlegel et al observe the same signal in their analysis.

L188/189: So is this attributable to the different datasets? Or something else?

L217/2018: Was there reason to believe that Schlegel's results strongly depend on the resolution of OISST? E.g. do important mechanisms that drive MCS occur on significantly smaller scales that could now be resolved with OSTIA? Is the most important difference of the dataset the resolution (even though OSTIA is also regridded) or the bulk vs. foundation temperature?

---

## Author Response (AR1)

**Point-by-point response to reviews 1 and 2**

I have put reviewer number ones comments into this document.

My responses are inline in **bold.**

**Reviewer 1 comments and replies.**

**Summary:**

The authors assess the intensity and frequency trends of marine heatwaves and marine cold spells over the satellite period, an analysis that will be included in the 7[th] edition of the Copernicus Marine Service Ocean State Report. The authors show that marine heatwaves have become more frequent and marine cold spells have become less frequent, findings that have been published before and agree with these previous studies. The authors also investigate changes in different event categories.

**Evaluation & Recommendation:**

The analysis is solid and the manuscript reads well. I only have a few minor points that needs to be addressed before I can recommend the paper for publication.

**Specific comments**

Title: Maybe clarify that you look at trends over the satellite period

**DONE. I've changed the title from "Comparing Global Trends in Marine Cold Spells and Marine Heatwaves" to "Comparing Global Trends in Marine Cold Spells and Marine Heatwaves using reprocessed satellite data"**

L7: Please write throughout the MS the two-word term "marine heatwave" instead of the three-word term "marine heat waves". This is the consensus amongst recent publications including IPCC reports. It also helps avoid confusion, probably amongst people outside of research, that this is not a literal 'wave' (e.g., wave on the beach), but rather an analogy to terrestrial heat waves (although here you also have a three-word term, but these waves are probably less prone to confusion).

**DONE**

L.8: Marine heatwaves or MCS do not have to be at sea surface, they also occur at depth. Change "sea surface temperature" to "ocean temperature". You can still say in the next paragraph that you used sea surface temperature to analyse MHWs and MCS at ocean surface.

**DONE**

L12: 'in large areas': Can you give a number in percentage of the global ocean here?

**This sentence refers to the trend analysis in figure 3. I apologise, but I have run out of time to code this calculation up and run it (it takes a large amount of time to perform its calculations).**

L15-16: Southern Ocean: I guess this is due to less warming of the SO surface. Maybe good to mention that.

**I've mentioned this - but in the bottom of Section 3.3.**

L17: Please clarify here that you mean the trend of increasing NUMBERs of MHWs. Because the fraction of area decreases (see Figure 4 right panel) for Moderate MHWs.

**DONE**

L41: Please cite here also the original references and not just the IPCC report. eg., Frölicher et al. 2018 (https://www.nature.com/articles/s41586-018-0383-9) and Oliver et al. 2018 (https://www.nature.com/articles/s41467-018-03732-9)

**DONE**

L43: Please include here Frölicher and Laufkötter (2018; https://www.nature.com/articles/s41467-018-03163-6)

**DONE**

L45: Please include here Cheung et al. 2021 when it comes to fisheries and socio-economic impacts (https://www.science.org/doi/10.1126/sciadv.abh0895)

**DONE**

L65: please include "e.g.," before the Schlegel reference, as there are many studies who have used the NOAA OISST record besides the Schlegel study.

**DONE**

L67: What is ref 2.1.1?

**DONE. "Ref 2.1.1" was the reference number of the OSTIA dataset that we used (see Table 1). This was added by Karina Von Schuckmann. I've replaced this with references to Table 1 so that it's less confusing.**

L 68: Please clarify what you mean with 'foundation SST product'.

**This has been clarified in the text.**

L74-75: Would it be possible to still use the 0.05° dataset for the two case studies described in section 3.1? I think the beauty of OSTIA lies in its very high spatial resolution.

**This would be possible but we preferred this case study to be consistent with the rest of the investigation.**

L78: Please acknowledge here that there are many other definitions that can be used or have been used in the literature besides Hobday.

**DONE**

78-102: Please clarify that you calculate all MHW/MCS characteristics at each individual grid cell separately and that you do not consider connected grid cells with MHWs or MCS, as was done for example in Laufkötter et al. (2020; https://www.science.org/doi/10.1126/science.aba0690)

**DONE**

Figure 1: Maybe highlight more clearly in the insets where the grid point is located. Maybe use red instead of white color.

**DONE**

Figure 1 caption: What do you mean with 98.625,0.875. Clarify which is the latitude and which is the longitude.

**DONE**

L126: I would change 'will help' to 'may help' as we do not know yet if that facilitates the communication. Or are there any examples that can be stated here, where the communication has improved because of the use of the category framework?

**DONE.**

Figure 2: It is difficult to see any color gradient in these figures. Please choose a better color scale to really highlight differences across regions. In general, the resolution of the figure could be improved.

L140-142: It is difficult to see these differences because of the chosen color scale.

**We've experimented with quite a few different colour scales (these changes were run past several reviewers) and this particular colourmap seemed to show up the best. As for the resolution – the original figures are actually high resolution, but you can't really see it because they have been shrunk to fit the page. I think that once the full-resolution originals have been uploaded, these should look a lot better.**

L143-148: I do not follow the explanation here. The Nino3 region also consistently experiences La Nina conditions with persistently low SSTs. The argument put forward here can also be applied to MCS. Or do I miss something? It seems to me that low temperatures are very variable – therefore more events per year, but probably shorter events, whereas high temperature are more persistent- therefore less events per year, but probably longer ones. But that is just a hypothesis.

**If this area of the ocean had high temperatures normally but strong dips with La Nina conditions, this would explain the low number of MHWs (less probability of exceeding average temp) and the high number of MCS (sharp dips below average). But this is really just a hypothesis.**

**So on further consideration, I think that this paragraph probably doesn't belong here because it is really explaining a consequence of the Hobday methodology (rather than giving an insight into what is actually happening in the Nino3 region – we don't know enough).**

**So I've decided to move this paragraph to the Hobday explanation section 2.2 to give a little more insight into the consequences of using climatology.**

L153: Maybe add here Frölicher et al. (2018; see link above) alongside the Oliver reference.

**I read the Frolicher paper and its topic is MHW climate forecasts. Whereas in this section we are discussing investigations comparable to this one. We have decided not to use the Frolicher paper here but have added the Frolicher reference in the other areas you suggested.**

L154: You may also state here that these earlier studies used the NOAA OISST.

**DONE**

Section 3.3 I guess most of the trends seen in MHWs and MCS (for the panels showing all events) can be explained by mean SST trends. The paper would therefore benefit of adding a mean SST trend figure somewhere in the manuscript. For example, the Southern Ocean and south of Greenland are both regions, where SST has not changed much over the satellite period (also the eastern equatorial Pacific) – therefore probably also the non-existing changes in MHWs and MCS.

**This would be very useful but we are limited to 4 figures in the report. Adding a mean SST trend figure would mean removing one of the other figures (which would make it harder to make our main point about the relative trends in MHW/MCS). So I can't do this, but I have added a paragraph to the end of Section 3.3 to talk about the literature around MHW being driven by SST rises.**

Figure 4: Maybe use the same y-axis scale for both panels for better comparison.

**It would definitely make comparison between MHW and MCS easier, but also make it harder to see the trends in the different categories for MHW (because they would be squashed together with a large axis). When we were generating the plots, we decided to focus on trends in the categories themselves (rather than a comparison).**

Section 3.4 Can you provide an explanation for the decrease in moderate MHW events over the satellite period?

**We don't know why that is – that is a very good question and an interesting topic for future research.**

L206: 'most of the changes are due to increases in category I MHWs'. Doesn't Figure 4 right panel (yellow line) show exactly the opposite?

**DONE. That was a mistake – thank you for finding it.**

L219: 'There are gaps': This comes a bit out of the blue. Can you elaborate a bit more here. What kind of gaps?

**I've changed this to be a bit more specific that it is the trends which are still a topic of research, I also moved this paragraph to an earlier point in the section to give a greater emphasis to our main point about the similarity of the results and that OSTIA can be future avenue of research. I have changed the ending of the report to talk more specifically about a follow-up using a higher-resolution OSTIA.**

L222: There are some studies out in the literature that have looked at predictability of MHWs (e.g. Jacox et al.; https://www.nature.com/articles/s41586-022-04573-9).

**Thank you for looking at this, but on further consideration, I have decided to remove this paragraph because it simply reiterates what has been said before about the Hobday framework in Section 3.1 and I think that it distracts from the main point that we are making in the conclusion (I have rewritten it to make it clearer based on Reviewer #2 comments).**

**Review: "Comparing Global Trends in Marine Cold Spells and Marine Heatwaves" by Robert Peal et al.**

This manuscript presents statistics and trends of cold extreme events, namely Marine Cold Spells (MCS), using the high-resolution CMEMS OSTIA sea surface temperature product, which is a foundation SST product, that is free of diurnal variability. The authors present two case-studies to demonstrate the application of the MCS detection algorithm. Then global comparison between MCS and Marine Heatwaves (MHW) statistics including trends is presented.

Ocean temperature extreme events, in particular MHWs, have received a lot of research attention over the past years, mainly due to their potentially devastating impacts on marine ecosystems as well as socio-economic impacts, through e.g. fisheries.

While the manuscript is logically structured, I have concerns as to the novelty of the results presented and recommend to re-consider publication of this manuscript after major revisions. Detailed comments are listed below.

**Main comments**

The authors claim that they are the first study to examine trends MCS and additionally comparing these to trends in MHWs. This is not true, as this study https://agupubs.onlinelibrary.wiley.com/doi/full/10.1029/2021GL097002, published in March 2022, does exactly that and the results presented by Peal et al in Fig.2 and 3 are basically identical with Fig.1 and Fig.2 in Wang et al., 2022.

The authors of the submitted manuscript do not cite this publication, which is a major flaw. I therefore recommend that the authors have to revise their manuscript significantly to demonstrate the novelty (if there is any) of their results.

**We are unclear of the basis of this comment, as we have not claimed in this paper to be the first people to examine trends in MCS and MHW (the question of trends has been investigated many times in the other papers that we have referenced in the report). The novelty of our research is that it is the first to use OSTIA for investigating MHW and MCS.**

**The Wang 2022 paper actually does claim in the conclusion to be the "first global study to quantify and compare the nature of multi-decadal trends (rates of change) in both MCSs and MHWs." Perhaps the reviewer got this report and the Wang paper mixed up?**

**The fact that our research uses OSTIA and produces results comparable with other research efforts using OISST is one of our key conclusions.**

**We make both these points in our abstract (lines 22-24). We reiterate our second point about the similarity of our results to other research efforts in the introduction (lines 53-60) and again in the conclusion (lines 212-215).**

**I have referenced the Wang 2022 paper in our report to add additional strength to our arguments.**

Furthermore, the authors mention that one novelty is using the CMEMS OSTIA dataset instead of OISST. While I see some value in comparing results using different datasets, I feel like the authors should then use chance to provide more quantitative details into the differences, e.g. what actual difference does it make for MHW/MCS statistics when I used OISST vs. OSTIA? How does this differ from region to region? This comes down to the drivers of the events, where do I really need such high resolution and where is 0.25° sufficient? Does OSTIA provide better results in coastal regions or areas of high SST gradients, such as Western Boundary Currents?

**These are very good questions which relate more specifically to a direct comparison between OSTIA and OISST, which would make a good topic for a research paper. We mentioned this in lines 216-218.**

**However, this is not the topic of the report, which is to evaluate trends in MHW and MCS using OSTIA (and in the conclusion we compare these results to similar ones found for OISST in the literature).**

Also the authors state that they are regridding OSTIA to 0.25° due to computational reasons. How does that affect the results? The authors could try to quantify this on a regional example if memory is an issue for the global dataset. I am not an R user but there should still be ways around these memory issues by e.g. applying the detection code on lat/lon chunks of the data.

**Unfortunately, the R programming language has a fundamental flaw in that it is very inefficient at storing data. Usually this problem isn't apparent, but it came up due to the size of our dataset. We tried various techniques. For example, we found that even reducing the lat/lon chunks still runs into problems because eventually, these chunks need to be combined into a global dataset, at which point, the memory overflows. Even running the code on a powerful cluster that we have here in the Met Office didn't mitigate this problem. This is why, in future investigations, we intend to use Python.**

In addition, highlight the differences of using a bulk vs. a foundation product? How does that impact my MHW/MCS statistics?

**The similarities between OISST and OSTIA mean it is unlikely to be a large issue, but it is a topic for future research.**

If this manuscript is intended to be more of a report on the current state, the authors should better discuss outstanding issues, challenges and also highlight potential drivers in their discussion. A main part though is being up to date on existing literature.

**Apologies for missing the recent Wang 2022 paper. This came about due to a delay between preparing the manuscript and when it could be submitted.**

**Minor comments**

There is little to no new insight provided as to the drivers of these events. If not through analyses, then the trend patterns should at least be discussed in the context of existing literature on temperature trends and their drivers.

**In lines 49-51 we discuss the current literature about drivers of MCS which is still an open question. However I have added a brief discussion of SST trends to the end of section 3.3.**

Section 3.4. provides some new insights but is again lacking some interpretation.

**This would be an interesting topic for future research.**

**Figures**

Fig 3: Why use such a pale colormap? Panels e)-j) are very hard to see. The similar comment applies to Fig2 panels c) and d). I believe using a sequential colormap with more colors would help the readability of the figures.

**We've experimented with quite a few different colour scales (these changes were run past several reviewers) and this particular colourmap seemed to show up the best. Part of the problem is that these plots are the low resolution versions. Once the full-resolution originals have been uploaded, these should look a lot better.**

Fig 4: I suggest making the lines a little thicker for better readability.

**DONE**

**Line-based comments**

L27: add 'can' after (MCSs), since not all events have devastating impacts

**DONE**

L42: same point as in L27. Change to for example 'These events often have dramatic…'

**DONE**

L47: Site https://www.science.org/doi/10.1126/science.abj3593 for socio-economic impacts. Change sentence to e.g. "… having socio-economic impacts (Smith et al., 2021) by e.g. altering fishery yields (Mills…)"

**DONE**

L78ff: Schlegel et al., 2021 should be clearly mention somewhere here.

**DONE**

L86: Why not use the term commonly used 'threshold'? The categories are then one,two,three,four times the threshold as also presented in Fig 1. This would seem more intuitive than presented here.

**DONE**

L93: Add that if there is a gap of less than two days between MHWs they are considered as one event.

**DONE**

L134ff: It would be more intuitive (from an oceanographer's point of view) to state at the beginning that Western Boundary Current regions show higher intensity and then list the regions.

**DONE**

L171-176: There are several ocean papers that discuss cooling trends over the Southern Ocean, these could be mentioned here to at least provide the reader with insights as to potential mechanisms behind these trends. Hence it is not surprising that Schlegel et al observe the same signal in their analysis.

**DONE. I have added a section on SST trends to the end of this section.**

L188/189: So is this attributable to the different datasets? Or something else?

**This is unknown. This would be an interesting topic for future research but as I said – this paper is about trends in OSTIA dataset, rather than comparing OSTIA to OISST.**

L217/2018: Was there reason to believe that Schlegel's results strongly depend on the resolution of OISST? E.g. do important mechanisms that drive MCS occur on significantly smaller scales that could now be resolved with OSTIA? Is the most important difference of the dataset the resolution (even though OSTIA is also regridded) or the bulk vs. foundation temperature?

**There was no real reason to think that the previous research findings were the result of the dataset, but it is good to confirm this independently because Schlegel 2021 talked about this at the end of section 4.2:**

**"Further investigation into how this result compares with different satellite products was beyond the scope of this study."**

**So this report is the beginning of answering this important question.**